# Dyggve–Melchior–Clausen Syndrome in Ecuador: Expanding Knowledge on a Rare Genetic Disorder

**DOI:** 10.3390/genes16050490

**Published:** 2025-04-25

**Authors:** Carlos Reyes-Silva, Joseline Gallardo-Vizuete, Judith Guzmán-Acán, Gabriela Jaramillo-Koupermann, Alejandro Cabrera-Andrade

**Affiliations:** 1Unidad de Genética, Hospital de Especialidades Eugenio Espejo, Quito 170403, Ecuador; carlosresil@yahoo.com; 2Posgrado de Endocrinología, Universidad de Las Américas, Quito 170125, Ecuador; joseline.gallardo@udla.edu.ec (J.G.-V.); judith.guzman@udla.edu.ec (J.G.-A.); 3Laboratorio de Biología Molecular, Subproceso de Anatomía Patológica, Hospital de Especialidades Eugenio Espejo, Quito 170403, Ecuador; gaby_jaramillok@yahoo.com; 4Grupo de Bio-Quimioinformática, Universidad de Las Américas, Quito 170125, Ecuador; 5Escuela de Enfermería, Facultad de Ciencias de la Salud, Universidad de Las Américas, Quito 170125, Ecuador

**Keywords:** Dyggve–Melchior–Clausen syndrome (DMC), *DYM* gene, dymeclin, c.1878delA, Mestizo population, Ecuador, rare genetic disorders

## Abstract

**Background**: Dyggve–Melchior–Clausen syndrome (DMC) is a rare autosomal recessive skeletal dysplasia characterized by dwarfism, coarse facial features, and intellectual disability. Caused by loss-of-function variants in the *DYM* gene, which encodes dymeclin, DMC is predominantly reported in consanguineous populations but remains poorly studied in South America. **Methods**: We report a 21-year-old Ecuadorian male with clinical features suggestive of DMC. Comprehensive clinical, radiological, and genetic evaluations were conducted, including clinical exome sequencing and Sanger sequencing, followed by an in silico analysis to assess the structural and functional consequences of the identified variant. **Results**: Exome sequencing identified a homozygous c.1878delA (p.Lys626fs) frameshift variant in the *DYM* gene, which was confirmed by Sanger sequencing as inherited from heterozygous parents. Variants of uncertain significance were detected in other skeletal dysplasia-related genes but did not correlate with the phenotype. A comprehensive review of reported *DYM* variants was also conducted. **Conclusions**: This report documents the first case of DMC in Ecuador and the second in South America, expanding the global understanding of DMC’s genetic diversity. It underscores the value of next-generation sequencing in rare disease diagnostics and highlights the critical need for inclusive genomic research in underrepresented populations to improve the understanding of genetic heterogeneity and rare disease epidemiology.

## 1. Introduction

Dyggve–Melchior–Clausen syndrome (DMC) is a rare autosomal recessive disorder classified under spondyloepimetaphyseal dysplasias, affecting the ossification process of the long bones’ epiphyses and diaphyses, as well as vertebral development. Its worldwide prevalence is estimated to be less than 1 per 1,000,000 people, with approximately 100 cases reported worldwide [1]. This condition manifests as short trunk dwarfism, distinctive coarse facial features, microcephaly, and intellectual disability [2]. Affected individuals exhibit a barrel-shaped thorax, brachydactyly, and knee malformations such as varus (bowlegged) and valgus (knock-kneed) deformities, alongside limited joint mobility, facial dysmorphisms, and micropenis. Radiological findings characteristic of DMC include platyspondyly with a double vertebral hump, small iliac wings with a lacy iliac crest, dysplastic acetabulum, epiphyseal dysplasia, shortened long bones with irregular metaphyses, and small hands [3]. Historically, DMC was categorized alongside inherited metabolic disorders due to its clinical overlap with conditions such as Morquio syndrome [4]. Despite sharing common clinical features with Morquio, DMC distinguishes itself by the absence of corneal opacity, deafness, valvular disease, and mucopolysacchariduria. Conversely, microcephaly and intellectual disability are hallmark characteristics consistently observed in DMC [5]. This distinction underscores the distinctive clinical presentation of DMC, setting it apart not only from metabolic disorders such as Morquio but also from other spondyloepimetaphyseal dysplasias, such as Smith–McCort dysplasia (SMC) [6].

The etiology of this syndrome stems from loss-of-function variants in dymeclin, encoded by the *DYM* gene located at chromosome 18q21.1 [7,8]. Dymeclin, a cytosolic protein that can be recruited to the Golgi apparatus, plays a crucial role in vesicular trafficking processes, including vesicle budding, transit, sorting, and membrane fusion [9]. It is essential for endochondral bone formation during early development, particularly affecting the processing, secretion, or uptake of growth factors, extracellular matrix components, and matrix remodeling proteases. Consequently, functional defects in dymeclin disrupt osteogenesis at multiple levels [10,11]. Furthermore, dymeclin’s significant contribution to early brain development suggests that its malfunction is associated with intellectual disability in affected individuals [12,13].

A significant portion of *DYM* gene variants in DMC cases is known to cause premature stop codons, resulting in the dymeclin protein’s loss of function [7,13,14,15,16,17,18]. Additionally, genetic variations, such as complex genomic duplication in exons 2 and 14 [15], along with splice acceptor variants located in introns 3, 4, 5, 7, 10, 11, 13, and 15 [13,17,18,19,20,21,22,23] and frameshift variants in exons 2, 8, 10, 11, 15, and 17 [7,13,18,20,22,24,25,26,27,28,29], have been documented. These findings highlight the array of molecular mechanisms that compromise dymeclin’s functionality, illustrating the complexity of genetic disruptions contributing to the pathology of this syndrome.

In this case report, we document the first Ecuadorian patient diagnosed with DMC. Utilizing comprehensive genetic analysis through massive sequencing, we identified the c.1878delA variant. This frameshift variant results in an aberrant elongation of the protein to 718 amino acids. Our findings underscore the critical role of advanced genetic testing, specifically massive sequencing, as a diagnostic tool for rare diseases. Moreover, this case contributes significantly to the understanding and characterization of rare genetic disorders within South American populations, highlighting the value of genetic diagnostics in diverse geographical contexts.

## 2. Case Presentation

### 2.1. Patient Presentation and Clinical Assessment

A 21-year-old male from Quito, Ecuador, was referred to the clinical genetics department at Hospital de Especialidades Eugenio Espejo at the age of 17 for the evaluation of suspected skeletal dysplasia. He was born at 38 weeks of gestation via uncomplicated cephalovaginal delivery, with a birth weight of 3180 grams and length of 47 cm. There was no known consanguinity or family history of genetic disorders, including DMC. During early childhood, developmental delays were noted, including independent sitting at 8 months, ambulation at 18 months with a lateral sway gait, and sphincter control at 3 years. Clinical concerns included short stature and abdominal distension, initially raising suspicion of mucopolysaccharidosis type VI (MPS VI).

At the time of referral (age 17), his height was measured at 137 cm (−3 standard deviations, −3 SD), and he weighed 42 kg, corresponding to a BMI of 22.4. He also presented with microcephaly (an occipitofrontal circumference of 45 cm). Physical examination revealed coarse facial features, macroglossia, mandibular prognathism, a short neck, a disproportionately short trunk, and pectus carinatum. Additional findings included spinal and hip deformities, thoracic kyphosis measuring 62 degrees, bilateral pes planus, upper limb deviations, and an unsteady gait with hip adduction and restricted movement (Figure 1). Cognitive evaluation using the Wechsler Adult Intelligence Scale (WAIS-IV) revealed an IQ of 68, consistent with mild intellectual disability, manifesting as restricted and repetitive language use. No formal body composition or functional handgrip assessments were performed. To ensure phenotypic precision, Human Phenotype Ontology (HPO) terms and corresponding codes for the clinical findings were documented (Appendix A).

### 2.2. Multidisciplinary Evaluation and Exclusion of MPS VI

The patient underwent multidisciplinary assessments, including cardiology, pulmonology, traumatology, endocrinology, and pediatric neurology. Spirometry revealed mild pulmonary restriction, with a forced vital capacity (FVC) of 0.93 L (58% predicted), a forced expiratory volume in one second (FEV_1_) of 0.93 L (57% predicted), a FEV_1_/FVC ratio of 100%, and a forced expiratory flow (FEF_25–75_) of 2.88 L (118% predicted). Despite residing in a high-altitude region (Quito, ~2850 m above sea level), the patient showed no clinical cyanosis or additional signs of hypoxia; however, arterial blood gases (pCO_2_/pO_2_) and hemoglobin levels were not obtained to further assess how altitude might compound the mild restrictive pattern. The cardiology evaluation indicated grade I cardiomegaly with a preserved left ventricular ejection fraction of 58%, and there were no clinical signs of heart failure. Given the initial suspicion of MPS VI, enzyme activity analysis of arylsulfatase B was performed (11.6 µmol/L/h; normal range ≥ 8.8 µmol/L/h) and found to be within normal limits, effectively ruling out MPS VI. Mild intellectual disability was also noted, consistent with the cognitive assessment findings described earlier.

### 2.3. Genetic Diagnosis

To identify potential genetic causes, clinical exome sequencing was performed using the Invitae Skeletal Disorders Panel (test code 89100, Invitae Corporation, San Francisco, CA, USA), which targets 358 genes implicated in skeletal dysplasias. Sequencing data were aligned to the GRCh38.p14 reference genome and analyzed using the Variant Effect Predictor (VEP) for functional annotation (https://www.ensembl.org/vep, accessed on 9 April 2024). Variants were interpreted with ClinVar (https://www.ncbi.nlm.nih.gov/clinvar/, accessed on 9 April 2024) and the Human Gene Mutation Database (HGMD) (https://www.hgmd.cf.ac.uk/ac/index.php, accessed on 12 April 2024). Additionally, variants were assessed using the guidelines of the American College of Medical Genetics and Genomics (ACMG) to classify them as pathogenic, likely pathogenic, or variants of uncertain significance, thereby aiding the diagnostic process. The analysis identified a homozygous frameshift variant in exon 17 of the *DYM* gene, NM_017653.6: c.1878del; p.(Lys626AsnfsTer94), which is consistent with the patient’s clinical phenotype and supports the diagnosis of DMC. This pathogenic variant met ACMG 2015 [30] criteria for classification as pathogenic (PM3, PVS1, PM2, and PP5). As expected for a recessive disorder, global population databases report a low but not absent allele frequency (gnomAD v4.1.0: 8 alleles in 34,586 individuals [0.02%], no homozygous cases). Additionally, ClinVar (Variation ID: 3191) contains an entry for this variant. Aside from the *DYM* pathogenic variant, variants of uncertain significance (VUS) were detected in *WISP3*, *CSPP1*, *GNPTAB*, and *FLNB*, genes known to be associated with skeletal dysplasias. However, these VUS were determined to have no significant contribution to the patient’s phenotype.

### 2.4. Validation by Sanger Sequencing

To confirm the findings and determine inheritance patterns, Sanger sequencing was performed on the proband, both parents, and the proband’s sister. Specific primers were designed (using the Primer-BLAST tool, https://www.ncbi.nlm.nih.gov/tools/primer-blast/, accessed on 6 June 2024) to amplify exon 17 of the *DYM* gene, and PCR amplification was performed using GoTaq^®^ Green Master Mix (Promega, Madison, WI, USA). The primers used for amplification were as follows: forward primer—5′-TTTCATGCCCTGCTGCTGTGACCAGTT-3′ and reverse primer—5′-AGTGTGCACTGTTGGATGGATGGATGGATAGTGT-3′. The results confirmed the autosomal recessive inheritance of c.1878delA: the proband was homozygous, while both parents and the sister were heterozygous carriers (Figure 2A,B), excluding the possibility of a de novo variant.

### 2.5. Dymeclin Functional Analysis

The c.1878delA variant introduces frameshift variants at codon 626, leading to the addition of 94 amino acids and an elongated dymeclin protein of 718 amino acids. As illustrated in Figure 2C, this variant is presented in the context of the *DYM* gene and the resultant protein, alongside a comparative overview of other coding-region variants reported worldwide in individuals diagnosed with DMC syndrome. Additionally, a Appendix A provides a comprehensive list of all variants reported in the *DYM* gene, including both coding-region and intronic variants identified in patients with DMC.

To further assess the impact of the c.1878del variant, an in silico analysis was performed using the bioinformatics platforms Swiss-Model and InterPro. Swiss-Model (https://swissmodel.expasy.org/, accessed on 7 August 2024) was used to predict the tertiary structure of the wild-type and mutant dymeclin proteins based on the NP_060123.3 reference sequence, with default settings applied for homology-based modeling. This approach enabled visualization of the structural alterations caused by the frameshift variant. InterPro (https://www.ebi.ac.uk/interpro/, accessed on 9 August 2024), a resource for classifying protein sequences into families and predicting domains, was employed to annotate both the reference and mutant proteins. It integrates predictive models from multiple databases to identify conserved regions and domain architecture [31]. In the reference sequence, InterPro predicted non-cytoplasmic, transmembrane, and cytoplasmic domains (Figure 2D). In contrast, the predicted mutant protein retained only the non-cytoplasmic domain, while transmembrane and cytoplasmic features were absent (Figure 2E), suggesting that the variant alters the protein’s three-dimensional conformation. In this context, “cytoplasmic” refers to regions predicted to face the cytosol, while “non-cytoplasmic” indicates segments oriented away from the cytosol, likely toward the lumen of intracellular organelles such as the Golgi apparatus. These findings indicate that c.1878delA disrupts dymeclin’s structural integrity, likely impairing its function.

### 2.6. Follow-Up and Clinical Management

The patient continues to receive multidisciplinary follow-up, particularly for mild tricuspid insufficiency and skeletal complications. Orthopedic concerns include increased dorsal kyphosis, bilateral coxarthrosis, reduced femorotibial interarticular spaces, and chronic bone pain. Supportive management focuses on symptom relief and mobility preservation, with ongoing monitoring to address potential complications.

## 3. Discussion

This report presents the first Ecuadorian patient diagnosed with DMC syndrome, highlighting the identification of the c.1878delA frameshift variant in the *DYM* gene. The patient’s clinical presentation aligns with previously reported phenotypic profiles attributed to *DYM* variants [21,24,25,29,32], which disrupt dymeclin function and result in skeletal anomalies such as vertebral deformities, kyphosis, scoliosis, and limb malformations, along with characteristic facial features, including a prominent forehead, short nose, and mandibular prognathism. Differentiating DMC from other skeletal dysplasias, particularly MPS VI, can be challenging due to overlapping skeletal manifestations; however, distinct clinical and laboratory findings, such as the absence of corneal opacity, normal urinary mucopolysaccharide excretion, and specific radiographic characteristics, aid in achieving an accurate diagnosis [17,22,33]. Although significant developmental delays and intellectual disability are uncommon in most skeletal dysplasias, cognitive impairment is a hallmark feature of DMC, and motor milestone delays often occur due to skeletal disproportions or joint hypermobility, further distinguishing it from other disorders.

Genetic alterations that impair dymeclin structure and function disrupt essential cellular processes, such as secretion and absorption, and have been implicated in both DMC syndrome and SMC dysplasia. While these syndromes share overlapping clinical features, intellectual disability is uniquely associated with DMC [6]. Dymeclin, an evolutionarily conserved protein consisting of 669 amino acids, lacks significant homologies or known functional domains, apart from the theoretical presence of an N-myristoylation site. It is predominantly expressed in osteoblasts, chondrocytes, and fetal brain cells, where it plays a critical role in regulating the Golgi apparatus-associated secretory pathway, essential for the development of endochondral bone formation and the initial stages of brain development [12,13,20]. As illustrated in Figure 2C, most variants associated with DMC are nonsense variants, which introduce a premature stop codon. Additionally, indels that cause reading frame shifts are reported to elongate the resulting protein. In the present case, we identified the c.1878delA (p.Lys626fs) variant in exon 17 of the *DYM* gene, which results in a frameshift at position 626 and extends the length of dymeclin by 94 amino acids, producing a 718-amino-acid protein. Our in silico analysis suggests that this variant lies within the region corresponding to the predicted cytoplasmic domain in the reference sequence, potentially compromising its aggregation capacity and overall functionality. Interestingly, when comparing the reference sequence (Figure 2D) to the mutated sequence (Figure 2E), the prediction for the mutant sequence no longer identifies transmembrane or cytoplasmic domains, suggesting a significant disruption in protein structure. Previous studies have demonstrated that nonsense variants, such as c.610C > T (p.Arg204Ter), c.1447C > T (p.Gln483Ter), and p.Lys616Ter, along with the missense variants c.1405A > T (p.Asn469Tyr), lead to the incorrect localization and aggregation of dymeclin. These variants cause intracellular accumulation of misfolded dymeclin, which forms ubiquitin-enriched cytoplasmic inclusions, ultimately leading to protein degradation [9]. Notably, these variants predominantly affect the predicted cytoplasmic domain, underscoring its critical role in dymeclin functionality. In contrast to nonsense variants, frameshift variants like c.1878delA, which alter the cytoplasmic domain by elongating the protein, emphasize the unique impact of this group of variants on dymeclin activity and its potential pathogenic consequences.

Despite the low prevalence of DMC worldwide, several studies have documented cases across diverse ethnicities and geographic locations, indicating that this syndrome does not exclusively affect specific ethnic groups. However, its autosomal recessive inheritance increases its prevalence in communities with a high rate of consanguineous marriages, which facilitate the manifestation of recessive diseases. A significant proportion of reported cases originates from families of Arab descent, particularly in countries such as Morocco, Lebanon, Tunisia, and Egypt [13,17,18,22,24,27,33], where a higher prevalence of *DYM* gene variants has been observed. Specifically, the c.1878delA variant, identified in our patient, is the most frequent variant in Moroccan populations, accounting for 54% of all variants reported in this group [13,17]. The case presented in this report is the first description of a patient with mestizo self-identification in Ecuador and marks the second case reported in South America [23]. This observation highlights the importance of adopting a broader perspective on genetic diversity in the study of rare diseases, as well as the need to extend epidemiological and genomic studies to underrepresented populations. These efforts not only improve the assessment of disease prevalence but also contribute to a deeper understanding of its genetic heterogeneity.

It is important to note that the pathogenic c.1878delA variant in homozygosity explains the clinical phenotype of the patient, supporting the diagnosis of DMC syndrome. However, additional VUS were identified in the *WISP3*, *CSPP1*, *GNPTAB*, and *FLNB* genes. These genes are associated with skeletal dysplasias, including progressive pseudorheumatoid dysplasia (*WISP3*), Joubert syndrome and ciliopathies (*CSPP1*), mucolipidosis types II/III (*GNPTAB*), and Larsen syndrome or other spondyloepiphyseal dysplasias (*FLNB*) [34]. While these disorders share clinical features with DMC, such as short stature, kyphosis, and vertebral flattening, there is currently no strong evidence to suggest that these VUS play a direct role in the observed phenotype. The identification of these additional variants highlights the inherent complexity of interpreting results from next-generation sequencing (NGS) in patients with rare diseases. As NGS frequently uncovers findings beyond the primary pathogenic variant, clinical correlation remains essential to avoid overinterpretation of uncertain results. Moving forward, further studies, including functional analyses and segregation testing in extended family members, may help clarify the potential contribution of these VUS. Additionally, re-evaluating these variants as new evidence emerges could provide insights into their significance, particularly in the context of modifier genes or dual diagnoses.

## 4. Conclusions

The description of a case of DMC syndrome in the Ecuadorian Mestizo population and the identification of the c.1878delA variant, previously reported in Moroccan groups, significantly expands our current understanding of the epidemiology of this syndrome. This finding highlights the importance of adopting a more inclusive and representative approach to the study of rare genetic diseases, acknowledging the intricate dynamics of genetic inheritance and the critical role of ethnic and geographic diversity in biomedical research. By broadening the scope of genomic studies to include underrepresented populations, this case contributes to a more comprehensive understanding of DMC and emphasizes the need for global perspectives in rare disease research.

## Figures and Tables

**Figure 1 genes-16-00490-f001:**
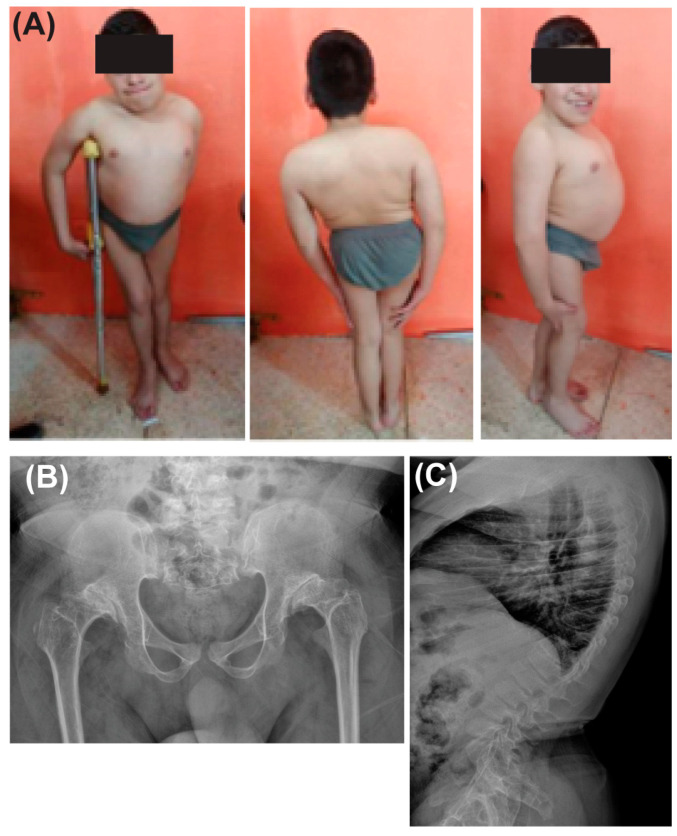
Clinical and radiographic findings of the patient diagnosed with Dyggve–Melchior–Clausen syndrome. (**A**) Frontal, posterior, and lateral views illustrating short stature (-3 SD); coarse facial features; short trunk; kyphosis; pectus carinatum; and hip deformities. (**B**) Pelvic X-ray showing acetabular dysplasia. (**C**) Lateral radiograph of the dorsal spine revealing platyspondyly.

**Figure 2 genes-16-00490-f002:**
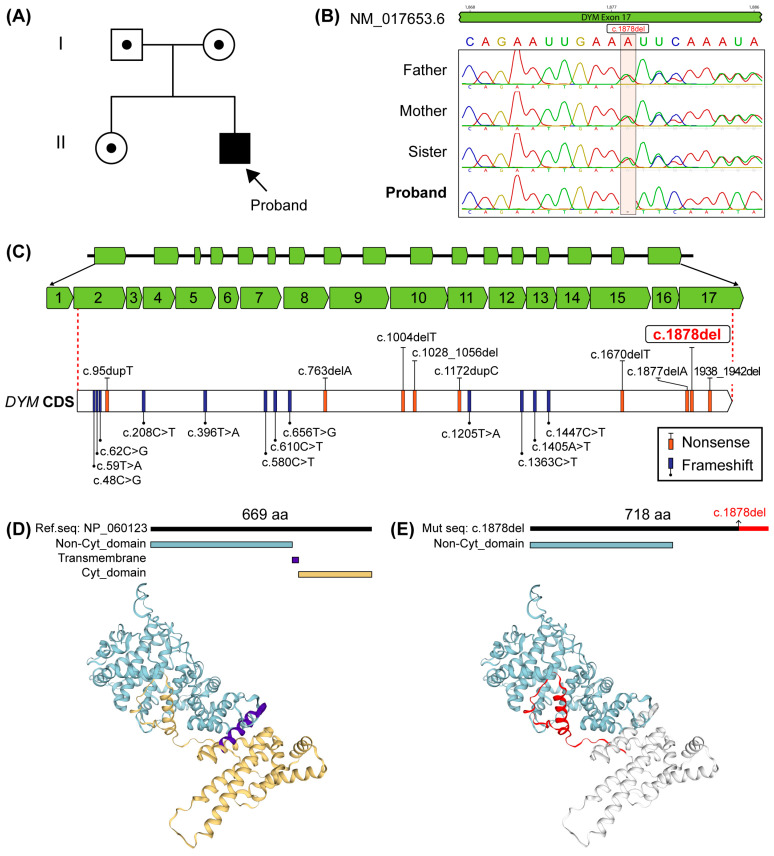
Family pedigree, genetic, and protein analysis of the reported case. (**A**) Pedigree illustrating the autosomal recessive inheritance pattern and the family members studied during the initial genetic evaluation. (**B**) Electropherogram showing the c.1878delA variants in *DYM*: the proband is homozygous for the deletion, whereas the father, mother, and sibling are heterozygous carriers. (**C**) Schematic representation of the *DYM* gene (green boxes denote exons). The transparent box highlights the coding region, showing the distribution of frameshift variants (orange boxes) and nonsense variants (purple boxes) previously reported worldwide in individuals with DMC syndrome. (**D**) Predicted functional domains of the *DYM* reference sequence. The in silico analysis identifies non-cytoplasmic (blue), transmembrane (purple), and cytoplasmic (yellow) domains. (**E**) DYM protein sequence harboring the c.1878delA variants. In the mutated sequence, only the non-cytoplasmic domain (blue) is predicted by InterPro. The remaining regions, which are not annotated as functional domains, are shown in gray. The segment highlighted in red represents the portion of the reference sequence that is lost as a result of the frameshift variants and subsequent premature termination. This model was generated using the mutant sequence to illustrate the predicted structural consequences of the c.1878delA variant.

## Data Availability

In compliance with local ethical guidelines, personal data cannot be disclosed; however, upon a justified request, the corresponding author may provide access to anonymized medical information.

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
