# Peer review of "Dyggve–Melchior–Clausen Syndrome in Ecuador: Expanding Knowledge on a Rare Genetic Disorder"

_genes, 2025, doi:10.3390/genes16050490_

Round 1

Reviewer 1 Report

Comments and Suggestions for Authors

The manuscript by Reyes-Silva et al. reports the first case of Dyggve-Melchior-Clausen (DMC) syndrome in Ecuador.  The authors present a careful clinical assessment of the patient that supports a diagnosis of DMC as opposed to other disorders such as Maroteaux–Lamy syndrome.  The authors describe a combination of exome and Sanger sequencing to demonstrate that heterogeneous parents with identical deletion mutations in the dymeclin (DYM) gene resulted in the recessive inheritance of the c.1878delA mutation.  The authors investigate the possible impact of the deletion mutation on the structure of the DYM protein using predictive programs at Expasy and InterPro.

The manuscript is well written and provides sufficient details for most of the experimental work.  There is also a thorough discussion of the global occurrence of mutations in DYM, clinical presentations, and structure/function consequences of these mutations. 

I have one criticism that should be addressed in revision.  There is no formal “Materials and Methods” section in the manuscript.  While, the genetic/sequencing analysis is described in sufficient detail, the structure modelling is not, and that leads to my confusion with Figure 2.  Points that need to be clarified:

  • Please describe in greater detail how Expasy and InterPro were used to generate the predicted structures.

  • Please define what is meant by “non-cytoplasmic” and “cytoplasmic” domains.  What organelle/cellular location is being referred to?

  • Line 159 “…the mutated sequence lacked transmembrane domains and retained only the cytoplasmic domain”; however, in the figure legend (line 173) it says “…the c.1878delT mutation. Only the non-cytoplasmic domain is predicted.”  This is confusing.  2E shows a structure for the cytoplasmic domain of the mutant.  The figure also shows the 94- amino acid extension resulting from the mutation.  How is this possible, if only the non-cytoplasmic domain was predicted?  While these structures are not critical to the paper, this portion of the manuscript needs to be improved, if these results are to be included.

Author Response

We thank the reviewer for his evaluation and comments. Suggestions to clarify the methodology and in silico strategies used are relevant to improve the presented case report. We have added a point-by-point response file.

Reviewer 2 Report

Comments and Suggestions for Authors

Dyggve-Melchior-Clausen evaluation

In this study the authors confirm the genetic diagnosis of a very uncommon disease, using the available methodology that allowed identification of the precise mutation of dymeclin involved in the development of the DMC syndrome. While genetical and biomolecular analyses were adequately described and referenced, case description lacks clinical information, as we comment below

1.Provide prevalence data

2.- Stature, BMI, microcephaly…: authors should provide the data supporting these features. From the illustrations, some dysmorphic traits are evident for the reader, and, moreover, the authors provide in Table S1 the HPO codes, but it is important to know the BMI, stature (what is below -3sD?), the data derived from pulmonary function assessment, ejection fraction, ecographically assessed myocardial thickness (are we dealing with a hypertrophic or dilated cardiomyopaythy?  Were there clinical signs of heart failure?), the severity of intellectual disability (any test performed?), ideally a body composition assessment if available, or functional assessment (handgrip?). Surely, at least some of these data can be easily collected by the authors. These data should be provided since they are essential for the clinical evaluation of this uncommon condition.

  1. Since some pulmonary restriction was appreciated, data about pC02, p02, hemoglobin should be added. Did the patient show cyanosis? If he lives in Quito or Ecuador highlands, breathing air with a low pO2, any superimposed (restrictive pattern in spirometry) increase of pCO2 may (severely) affect alveolar oxygen partial pressure. A table with some biological data may be of interest to the reader.
  2. Which are the levels of arylsulfatase (mentioned in the text)?

Other minor points

Line 187: it is stated “small jaw”. This is in contrast with what we can guess from figure 1, and what is written in line 93, and with the case descriptions by other researchers (for instance, Beighton, 1990 “bulky jaws”; Aglan et al., 2009; Obara et al., 2022,…). Please clarify.

Line 231: perhaps Egypt could be added ?( Aglan et al., 2009, cited by the authors, reference 30)

Author Response

We thank the reviewer for all the suggested comments, which helped to improve the clinical details
of the case report. Based on these suggestions, we added anthropometric data, spirometry values, IQ
assessment results, and clarified the context of high-altitude residence. We also updated the
geographic reference to include Egypt, as recommended. We have added a point-by-point response file.

Round 2

Reviewer 2 Report

Comments and Suggestions for Authors

The authors have provided most of the data required, so I have no further comments 

Author Response

We thank the reviewer for the evaluation and confirmation.